# Atomic resolution structures of the methane-activating enzyme in anaerobic methanotrophy reveal extensive post-translational modifications

Marie-C. Müller[1], Martijn Wissink [2], Priyadarshini Mukherjee[1], Nicole Von Possel [1], Rafael Laso-Pérez [3], Sylvain Engilberge[4], Philippe Carpentier[5,6], Jörg Kahnt[7], Gunter Wegener [1,8], Cornelia U. Welte [2] & Tristan Wagner [1,4] ✉

Anaerobic methanotrophic archaea (ANME) are crucial to planetary carbon cycling. They oxidise methane in anoxic niches by transferring electrons to nitrate, metal oxides, or sulfate-reducing bacteria. No ANMEs have been isolated, hampering the biochemical investigation of anaerobic methane oxidation. Here, we obtained the true atomic resolution structure of their methane-capturing system (Methyl-Coenzyme M Reductase, MCR), circumventing the isolation barrier by exploiting microbial enrichments of freshwater nitrate-reducing ANME-2d grown in bioreactors, and marine ANME-2c in syntrophy with bacterial partners. Despite their physiological differences, these ANMEs have extremely conserved MCR structures, similar to homologs from methanogenic *Methanosarcinales*, rather than the phylogenetically distant MCR of ANME-1 isolated from Black Sea mats. The three studied enzymes have seven post-translational modifications, among them was a novel 3(*S*)-methylhistidine on the γ-chain of both ANME-2d MCRs. Labelling with gaseous krypton did not reveal any internal channels that would facilitate alkane diffusion to the active site, as observed in the ethane-specialised enzyme. Based on our data, the methanotrophic MCRs should follow the same radical reaction mechanism proposed for the methane-generating homologues. The described pattern of post-translational modifications underscores the importance of native purification as a powerful approach to discovering intrinsic enzymatic features in non-isolated microorganisms existing in nature.

In light of climate change, studies about planetary methane fluxes have become increasingly important due to the global impact of the greenhouse gas. In anoxic marine sediments, about 71% of biogenic methane is transformed into the less potent greenhouse gas $CO_2$ through the process of anaerobic oxidation of methane (AOM)[1]. The same is true in terrestrial and freshwater systems, where AOM may account for up to a 50% reduction in methane emissions in freshwater wetlands[2]. The only microorganisms capable of performing AOM without the use of oxygen as a methane-activating agent are anaerobic methanotrophic archaea (ANME)[3,4]. All ANME groups have been found

in marine sediments except ANME-2d[5,6]. Electrons retrieved during the complete oxidation of methane can be transferred to sulfate-reducing partner bacteria (e.g. *Desulfobacterota*), most likely through direct interspecies electron transfer via nanowires[7,8]. Reports proposed that some ANMEs performed non-syntrophic AOM by coupling methane oxidation to Fe(III), Mn(IV), or humic substances reduction[9–12]. In addition, freshwater ANME-2d can reduce nitrate[13] or colonise electrodes and generate electricity[14].

Despite their common ability to turn methane into $CO_2$, ANME-1 and ANME-2/ANME-3 are phylogenetically distant. To understand their physiological differences, it is of utter importance to investigate the fundamental molecular mechanisms of AOM. Despite extensive efforts, the gathered biochemical information on ANMEs is scarce due to the absence of any axenic cultures. However, some laboratories have succeeded in obtaining enrichments of microbial consortia[15–17], a strenuous process due to the extremely long doubling times of ANME. This achievement can represent a turning point in deciphering how these microbes catabolise methane and might inspire biotechnological applications.

Methane capture, the first reaction in AOM, has been proposed to be carried out by the Methyl-Coenzyme M Reductase (MCR). Accordingly, the reaction would follow the reverse direction of methanogenesis[18]. In the proposed mechanism, the heterodisulfide, made of Coenzyme M (HS-CoM) and Coenzyme B (CoB-SH), would react with methane to generate methyl-S-CoM and free CoB-SH through a radical mechanism catalysed by the Ni-containing tetrapyrrole cofactor $F_{430}$[19,20]. Methane conversion to methyl-S-CoM was previously proved in the methanogenic enzyme from *Methanothermobacter marburgensis*[21] despite its unfavourable thermodynamics (Gibbs free energy change of +30 kJ mol$^{-1}$ of transformed methane)[4]. To perform this difficult process, ANMEs produce considerable quantities of MCR, which is by far the most abundant cellular enzyme[18,22,23]. In addition, a thermodynamic pull allowing an immediate conversion of methyl-S-CoM should proceed, leading to subsequent transformation to $CO_2$.

MCR quaternary structure consists of a dimer of heterotrimers $(\alpha\beta\gamma)_2$ harbouring the cofactor $F_{430}$ in each active site[20]. The catalytic chamber and its close surroundings contain many post-translational modifications (PTMs) that vary among different methanogens and are located on the α-subunit. No correlation has been observed so far between the PTMs and phylogeny. While for some PTMs the installation machinery has been resolved, their physiological functions remain elusive[24–27]. The structures of ten methanogenic MCRs have yielded considerable information, highlighting conserved overall features[19,28]. In contrast, the MCR structure from ANME-1 revealed unique traits, including a modified methylthio-$F_{430}$ cofactor, a cysteine-rich patch, and two different PTMs (i.e. a C7-hydroxy-tryptophan and a methionine sulfoxide)[29], corroborated by biochemical studies[18,22].

The unique traits hint towards the presence of other unexpected features in ANME MCRs. Acquiring snapshots of the phylogenetically different counterparts in ANME-2 and ANME-3 would stimulate applied research in methane mitigation. Attempts have been made to facilitate structural studies of ANME MCRs, including successful recombinant expression of ANME-1 MCR and the Ethyl-CoM reductase (ECR) specialised to activate ethane from '*Candidatus* Ethanoperedens thermophilum'[30]. However, this approach is biased as the machinery for generating and installing native $F_{430}$ variants, species-specific PTMs, and dedicated chaperones required for correct folding are absent in the recombinant host. Moreover, artefactual chimeric assemblies have been observed between the native MCR from the methanogenic host and the recombinant ANME-2 MCR-γ. Similarly, the cultivation-independent approach of computational protein structure modelling exhibits biases as the process is guided by extracted data from known models, but cannot predict the physiological cofactor, PTMs, and other unknown important functional traits such as water

networks and gas channels. As an illustration, the native ECR harbours a dimethylated-$F_{430}$ cofactor, a widened active site and unique PTMs ($N^2$-methylhistidine, a S-methylcysteine at a position different to the methanogenic one, and 3-methylisoleucine) that scaffold a gas channel connecting the active site to the solvent, favouring alkane diffusion[23]. Therefore, to explore the specific features of alkane-capturing enzymes, the investigation of native systems still prevails, which is a challenge in the case of non-isolated microbes such as ANMEs.

In this study, we purified MCRs directly from microbial enrichments dominated by ANME-2 to compare them with methanogenic and methanotrophic (ANME-1) homologues. The native structures obtained at atomic resolution reveal the detailed architecture of the methane-capturing enzyme, with the discovery of a novel PTM on the γ-subunit. These findings expand our knowledge of this crucial carbon-cycling enzyme, highlighting the importance of studying natural systems.

## Results

### Isolation of the core catabolic enzyme MCR from freshwater and marine ANME-2

Cultures containing an abundant population of ANME are rare due to the laborious enrichment procedures and the long doubling time of these organisms. Successive breakthroughs in the cultivation of ANME-2 species have enabled us to access freshwater and marine representatives for studying differences between their MCRs through native protein purification (Fig. 1 and Supplementary Fig. S1). In all presented cases, the native protein isolation was based on MCR tracking by following its intrinsic properties: the absorption signature from the $F_{430}$ cofactor and the typical pattern on denaturing gels (SDS-PAGE, Supplementary Fig. S2).

Investigations were started on a relatively simplified system in which ANME-2d grew independent of sulfate-reducing partners. Biological samples were retrieved from Ooijpolder (Netherlands)[31,32] and Vercelli rice fields (Italy)[14,17] and cultivated in bioreactors for >5 years, significantly enriching ANME-2d with nitrate as an electron acceptor. Metagenomic analysis of the Ooijpolder samples directly obtained during sampling time showed a read-based relative abundance of 28% '*Ca*. Methanoperedens sp.' BLZ2 (Supplementary Fig. S1)[33] (further referred to as ANME-2d$^O$). In comparison, metagenomic analysis of the Vercelli rice field enrichment sample showed a read-based relative abundance of 9% '*Ca*. Methanoperedens' (Supplementary Fig. S1), with the main strain '*Ca*. Methanoperedens' Vercelli Strain 1[34] (further referred to as ANME-2d$^V$). Both samples were separately processed for protein purification, starting from 17.5 g and 12.5 g of ANME-2d$^O$ and ANME-2d$^V$ containing biomass and yielding 114 mg and 14 mg of the major MCR, respectively (Fig. 1).

Furthermore, the purification setup was tested on a microbial enrichment from a marine system. Here, the original sample used to initiate the enrichment was collected from the Amon Mud Volcano in the Mediterranean Sea at a depth of 1120 m and a temperature of 14 °C. The microbial enrichment, grown at 20 °C in batch serum bottles since 2003, mainly gathered ANME-2c (with a relative abundance of 32%) along with microbial partners based on 16S rRNA gene sequencing[7] (Supplementary Fig. S1). The processed biomass of 3.21 g contained the microbial mixture as well as inorganic particles, and a final quantity of 4.9 mg of MCR could be extracted.

In all cases, the final purified MCR fractions subjected to UV/Vis spectroscopy presented a peak at 425 nm due to the $F_{430}$ cofactor in accordance with previous reports for methanogenic MCRs (Supplementary Fig. S2d)[23,35]. While the final SDS-PAGE suggests a high sample homogeneity in solution, we could not exclude a mixture of MCRs from different subspecies. Therefore, we used the selective crystallisation process to isolate and identify a single variant by X-ray crystallography.

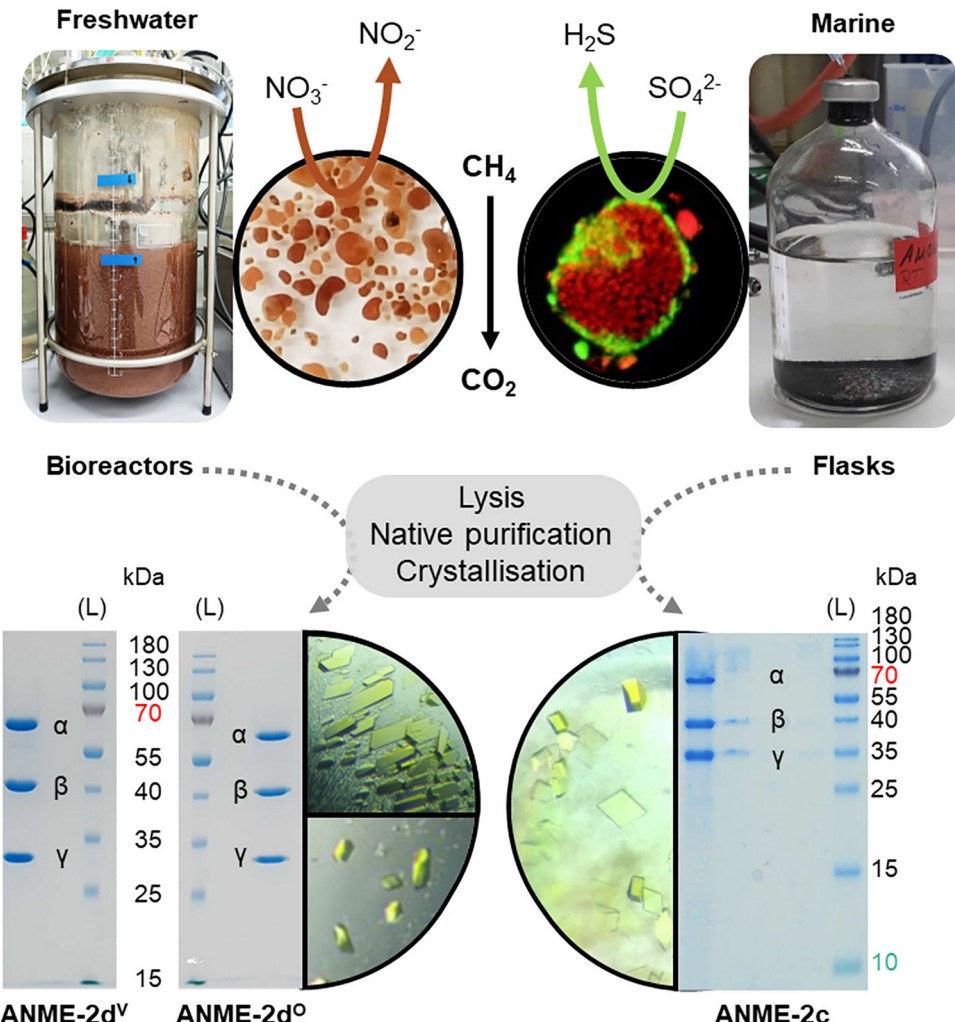

**Fig. 1 | Comparison of the ANME-2 samples investigated in this study and the procedure of protein isolation by native purification and crystallisation.** Images of the bioreactor and flasks containing the respective freshwater and marine ANME-2 are presented. Close-ups of granular biomass of ANME-2d reducing nitrate and fluorescence in situ hybridisation labelled micrographs[7] of the marine methanotrophic consortium (red for the archaeon and green for the sulfate-reducing partner, using specific probes) are shown. Purified MCR fractions are shown on the SDS-PAGE at the bottom of the figure next to the crystals obtained from each sample (top being MCR ANME-2d$^V$ and bottom ANME-2d$^O$). For ANME-2c, the sample passed on SDS-PAGE corresponds to dissolved crystals, and a picture of the purification procedure can be examined in Supplementary Fig. S2. L stands for ladder. Purification procedures were performed once for ANME-2c and ANME-2d$^V$ and three times for ANME-2d$^O$.

## MCR shows a well-conserved overall architecture

X-ray crystal structures were solved for MCR from ANME-2d$^O$ and ANME-2d$^V$, and models were refined to a final resolution of 0.98 Å (Table 1). Both MCRs crystallised in a monoclinic space group with similar unit cell dimensions. Despite the differences in the organisms and the slight deviations in the crystallisation solution (see section 'Methods'), both MCR models shared the same crystalline packing (Supplementary Fig. S3). The observation suggests a very close sequence identity that would maintain the surface contacts to establish the same crystal lattice. This was confirmed after sequencing the MCR from the initial electron density, matching the sequences derived from metagenomic data. MCRs from ANME-2d$^O$ and ANME-2d$^V$ are structurally almost identical with a root mean square deviation of 0.09 Å (with an overall of 2186 aligned Cα), probably due to their high sequence identity (with an identity of 95.4%, 97.2% and 95.6% for α, β, and γ, respectively),

The dataset obtained for the ANME-2c enrichment was also of sufficiently high quality to unequivocally determine the sequence from the electron density map at a resolution of 1.34 Å. The derived sequence perfectly matched the one of ANME-2c MCR from the metagenomic data (Supplementary Figs. S4 and S5). Based on the crystal structure, MCR from ANME-2c seems to form stable dimers (Supplementary Fig. S3), as pointed out by the PISA server analyses (the dimer exhibiting a $\Delta G^{int}$ of −617.2 kcal mol$^{-1}$ and a $\Delta G^{diss}$ of 26.8 kcal mol$^{-1}$. Analyses performed on the 25th of November 2024). The dimeric interface is exclusively formed by the interaction of the α-subunits, mainly via the C-terminal extension. The dimeric state does not disturb the access to the active site. Due to the low quantities of MCR obtained at the end of the purification and the slow growth of the consortium, we were unable to test whether this dimeric arrangement reflects a crystallisation artefact or a stable entity in solution. Further studies will need to be carried out. On the other hand, the predicted trimeric arrangement of the MCR from ANME-2d$^O$ observed in blue native PAGE[33] was not detected in the crystal structure. Nevertheless, we cannot exclude that the purification and crystallisation process might have disrupted a trimeric organisation occurring in vivo.

Despite the different environmental niches, the marine ANME-2c and freshwater ANME-2d MCRs have high amino acid sequence identities of 63.6%, 61.5%, and 71.4% for α, β, and γ subunit, respectively (Supplementary Figs. S5 and S6). The overall structure of the three

**Table 1 | X-ray analysis statistics for ANME-2 MCR**

| | ANME-2c from microbial enrichment | ANME-2c (Kr pressurised) | ANME-2d$^O$ from bioreactor | ANME-2d$^V$ from bioreactor |
|---|---|---|---|---|
| **Data collection** | | | | |
| Synchrotron and beamline | SOLEIL, PROXIMA-1 | ESRF, ID23-1 | ESRF, BM07-FIP2 | ESRF, BM07-FIP2 |
| Wavelength (Å) | 0.97856 | 0.86100 | 0.97980 | 0.97930 |
| Space group | $P2_1$ | $P2_12_12_1$ | $P2_1$ | $P2_1$ |
| Resolution (Å) | 127.16–1.34 (1.50–1.34) | 59.40–1.80 (1.83–1.80) | 76.67–0.98 (1.04–0.98) | 94.91–0.98 (1.04–0.98) |
| Cell dimensions | | | | |
| $a$, $b$, $c$ (Å) | 156.87, 157.46, 215.42 | 153.48, 153.67, 212.88 | 81.58, 189.30, 84.11 | 81.62, 189.82, 83.93 |
| $\alpha$, $\beta$, $\gamma$ (°) | 90, 90.34, 90 | 90, 90, 90 | 90, 114.27, 90 | 90, 114.36, 90 |
| $R_{merge}$ (%)[a] | 16.8 (167.1) | 12.7 (162.3) | 11.9 (116.8) | 7.9 (80.0) |
| $R_{pim}$ (%)[a] | 7.3 (73.7) | 5.1 (64.6) | 4.5 (46.8) | 3.0 (32.0) |
| $CC_{1/2}$ [a] | 0.991 (0.560) | 0.999 (0.768) | 0.996 (0.667) | 0.998 (0.754) |
| $I/\sigma_I$[a] | 6.6 (1.6) | 14.1 (1.8) | 9.4 (1.7) | 13.0 (1.9) |
| Spherical completeness[a] | 66.2 (12.2) | 100.0 (100.0) | 84.4 (24.4) | 74.7 (23.5) |
| Ellipsoidal completeness[a] | 96.2 (66.6) | / | 96.5 (70.9) | 83.9 (62.6) |
| Redundancy[a] | 6.3 (6.0) | 13.8 (14.2) | 7.8 (7.1) | 7.6 (7.0) |
| Nr. unique reflections[a] | 1,527,343 (76,367) | 461,593 (22,707) | 1,130,261 (56,513) | 996,827 (49,841) |
| **Refinement** | | | | |
| Resolution (Å) | 98.93–1.34 | 54.30–1.80 | 41.78–0.98 | 41.71–0.98 |
| Number of reflections | 1,527,210 | 461,298 | 1,130,120 | 988,481 |
| $R_{work}/R_{free}$[b] (%) | 11.24/14.71 | 12.93/15.52 | 10.25/11.76 | 12.96/14.81 |
| Number of atoms | | | | |
| Protein | 152,558 | 75,375 | 39,210 | 39,298 |
| Ligands/ions | 1328 | 1000 | 406 | 473 |
| Solvent | 9812 | 4293 | 3419 | 3293 |
| Mean $B$-value (Å$^2$) | 15.77 | 29.30 | 10.72 | 10.03 |
| Molprobity clash score | 0.99 | 1.50 | 0.60 | 0.45 |
| Ramachandran plot | | | | |
| Favoured regions (%) | 97.33 | 97.11 | 97.16 | 97.08 |
| Outlier regions (%) | 0.07 | 0.08 | 0.08 | 0.08 |
| rmsd[c] bond lengths (Å) | 0.010 | 0.009 | 0.010 | 0.011 |
| rmsd[c] bond angles (°) | 1.247 | 1.170 | 1.341 | 1.382 |
| **PDB code** | 9QR3 | 9QM5 | 9QR1 | 9QQT |

[a]Values relative to the highest resolution shell are within parentheses.
[b]$R_{free}$ was calculated as the $R_{work}$ for 5% of the reflections that were not included in the refinement.
[c]rmsd root mean square deviation.

ANME-2 MCRs matches that of the different methanogenic MCRs, of the methanotrophic ANME-1 MCR, and of the ECR from '*Ca*. E. thermophilum' (Fig. 2, Supplementary Figs. S5 and S6), the closest homologue being the MCR from *Methermicoccus shengliensis*[35]. Besides the overall similarity, notable structural differences are summarised in Fig. 2. The electrostatic profile of the surface is also similar to previously studied MCRs, with both ANME-2d members exhibiting more pronounced positive charges at the entrance leading to the active site than a typical methanogenic MCR (Supplementary Fig. S7).

**A sealed, strictly conserved catalytic chamber covered by seven post-translational modifications**
The active site of the three ANME-2 MCRs is highly conserved across all described MCRs from methanogenic archaea and contains a canonical $F_{430}$ cofactor in accordance with previous studies[22] (Fig. 3, Supplementary Fig. S8). The subatomic resolution of both ANME-2d structures enables the visualisation of the majority of hydrogen atoms and accurately depicts the position of each ligand atom. MCRs from ANME-2c and ANME-2d$^O$ contain reduced coenzymes trapped in the active site (i.e. CoM-SH and CoB-SH). However, MCR from ANME-2d$^V$ exhibits a mixture of ligands. The fit to the electron density was optimal when the model included CoM-SH, CoB-SH, the heterodisulfide, and methyl-

CoM (Supplementary Fig. S9). The methyl-CoM, modelled for the first time in an experimental structure, fits a position proposed in a recent molecular dynamics simulation model by Polêto and colleagues with a S–Ni distance of 3.9 Å[36]. However, due to its low occupancy and the requirement for experimental validation to confirm its existence, we omitted the methyl-CoM from the final deposited model.

The ligand-coordinating residues are almost perfectly conserved (Fig. 3a, b, Supplementary Fig. S10). In both ANME-2d, a canonical glutamine (position α341 in ANME-2c), proposed to be important to anchor the $F_{430}$, and juxtaposed to the tyrosine coordinating the CoM thiol (position α342 in ANME-2c), is substituted by a methionine (positions α343 and α347 in ANME-2d$^O$ and ANME-2d$^V$, respectively)[36]. The substitution does not impact the $F_{430}$ position and provokes few steric displacements in the vicinity (Supplementary Fig. S11).

The discovery of a gas channel in the ethane-capturing enzyme prompted us to investigate whether ANME-2 MCR possesses a similar access system for methane. Analysis of the internal cavities in all three models, using CAVER software[37] with a probe radius similar to that of methane, did not reveal tunnels protruding from the active site. In comparison, applying the same parameters to the ECR model from '*Ca*. E. thermophilum' revealed its hydrophobic substrate tunnel. Still, we performed krypton labelling on ANME-2c MCR crystals to verify the

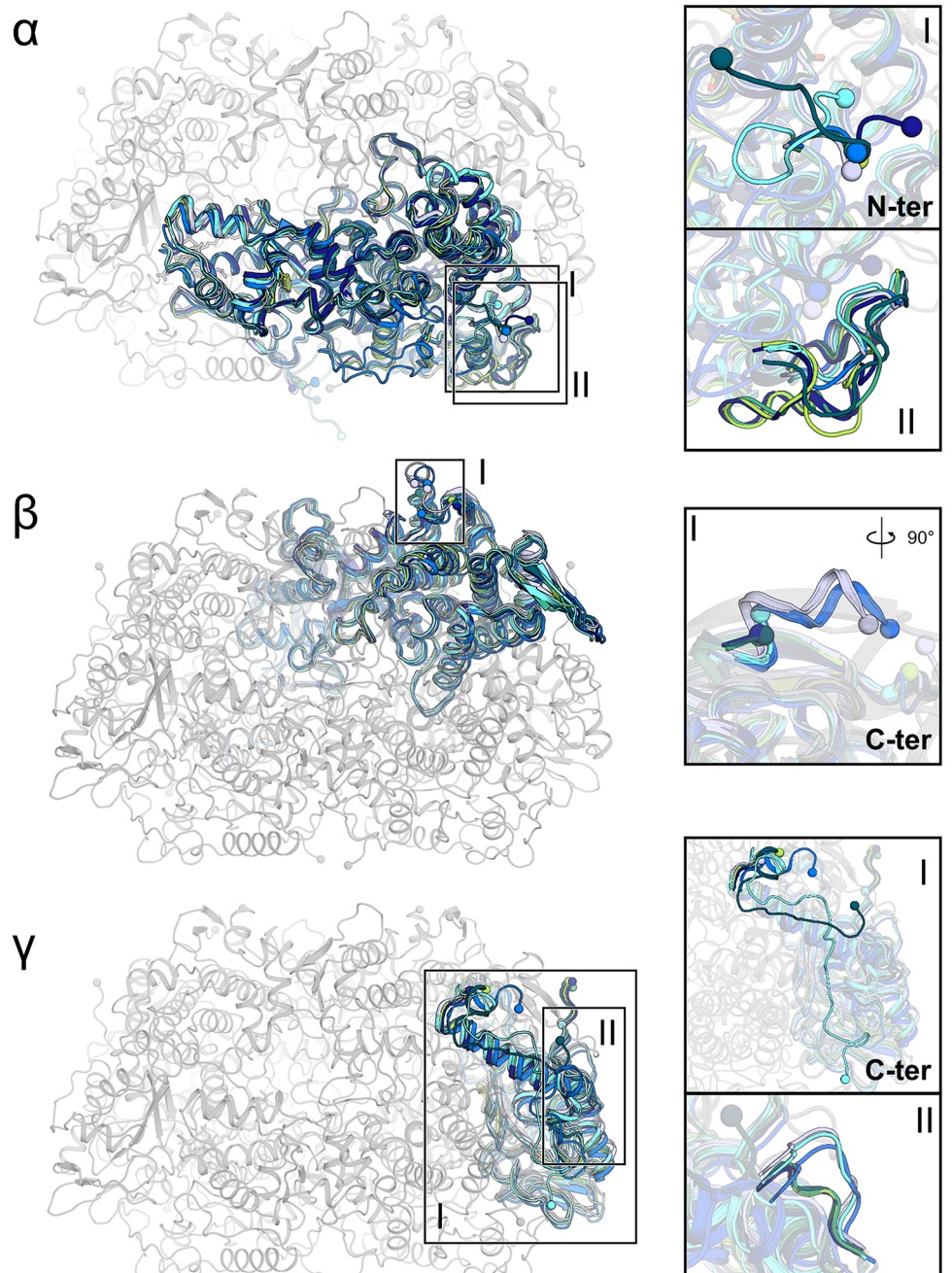

**Fig. 2 | Comparison of ANME-2 MCR structures with previously structurally characterised MCRs.** Superposition of the α-, β-, and γ-subunits of MCR. The panels on the right are close-ups of structural deviations. In each panel, the corresponding chain is coloured, while the remaining chains are shown (greyed out) for ANME-2d^V. MCR structures are coloured according to the microorganism, with *M. marburgensis* (PDB code: 5AOY) in light blue, *M. shengliensis* (7NKG) in deep blue, ANME-1 from Black Sea mats (3SQG) in cyan, '*Ca*. E. thermophilum' ECR (7BIS) in marine, ANME-2c in dark green, ANME-2d^O in green and ANME-2d^V in light green. N- and C-termini are shown as spheres.

existence of an accessory tunnel via experimental means. Based on the anomalous signal of the dataset collected at the krypton K-edge, no signals were observed in the active site or tracking in a hydrophobic tunnel (Fig. 3c, Supplementary Fig. S12). Krypton atoms are rather positioned on the outer shell of the protein, trapped in hydrophobic pockets, as previously observed for the methanogenic enzyme from *M. marburgensis*[23,38]. The absence of a dedicated gas tunnel connecting the active site to the solvent reinforces the proposed scenario that ethane-specialised homologues evolved this strategy by introducing loop extensions and dedicated PTMs. Enzymes targeting longer alkanes should also have a dedicated entry cavity[19].

All three ANME-2 MCRs contained a notably high number of seven PTMs based on the electron density model (Fig. 4, Supplementary Figs. S13 and S14). ANME-2c MCR includes the conserved $N^1$-methyl-histidine and thioglycine, the common 5(*S*)-methylarginine, 2(*S*)-methylglutamine, and the rarer S-methylcysteine, didehydroaspartate, and 6-hydroxytryptophan. The same PTMs are present in both ANME-2d MCRs, except for 2(*S*)-methylglutamine. Instead, the histidine 159 from the γ chain harbours a methyl group on its β-carbon in an *S* configuration (i.e. 3(*S*)-methylhistidine). The histidine is canonical and positioned at a distance of 3.5 Å from the $F_{430}$ cofactor, with the atom ND1 forming a hydrogen bond to $F_{430}$. The modified methyl group is

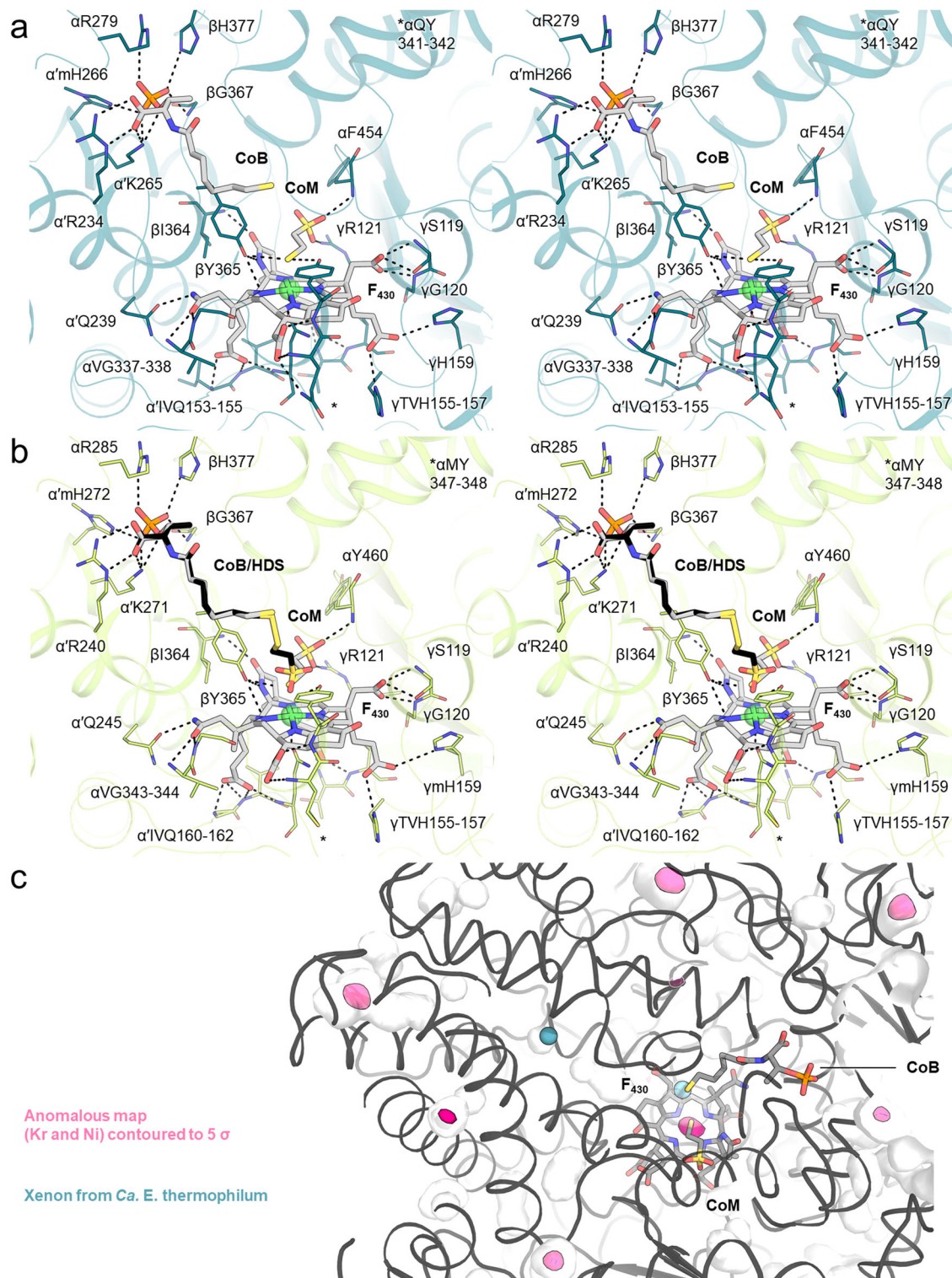

**Fig. 3 | ANME-2 MCR active sites.** Stereo-view of the MCR active site in **a** ANME-2c and **b** ANME-2d[v]. Interacting residues are shown as lines, and non-interacting main chains were omitted. Interacting residues are labelled as follows: chain-one letter amino acid code-residue number. Polar contacts are in black dashes. **c** Anomalous map highlights Kr positions (pink surface). Xe (teal spheres) from 'Ca. E. thermophilum' (extracted from PDB 7B2C) are superposed. Proteins are in cartoons, and ligands in sticks coloured as follows: red, blue, orange and yellow, for O, N, P and S, respectively.

distanced at ~5.7 Å and directed away from the cofactor without disturbing the positioning of surrounding residues (Supplementary Fig. S15). Upon careful analysis of $F_{430}$ superposition based on atomic resolution structures (Supplementary Fig. S15), we noticed that the methylhistidine slightly repositions the close-by $F_{430}$ propionate group in front of an axial histidine, optimising the contact without

modifying the salt bridge distance (2.9 Å). It must be noted that the modified histidine 159 is located on a loop that would be exposed to the solvent on the γ-subunit alone. Therefore, while histidine 159 would be hindered in the MCR complex, it would be accessible to a dedicated methyl-transferase for the PTM installation when the γ-subunit is nascent from the ribosome. Besides their clear presence in

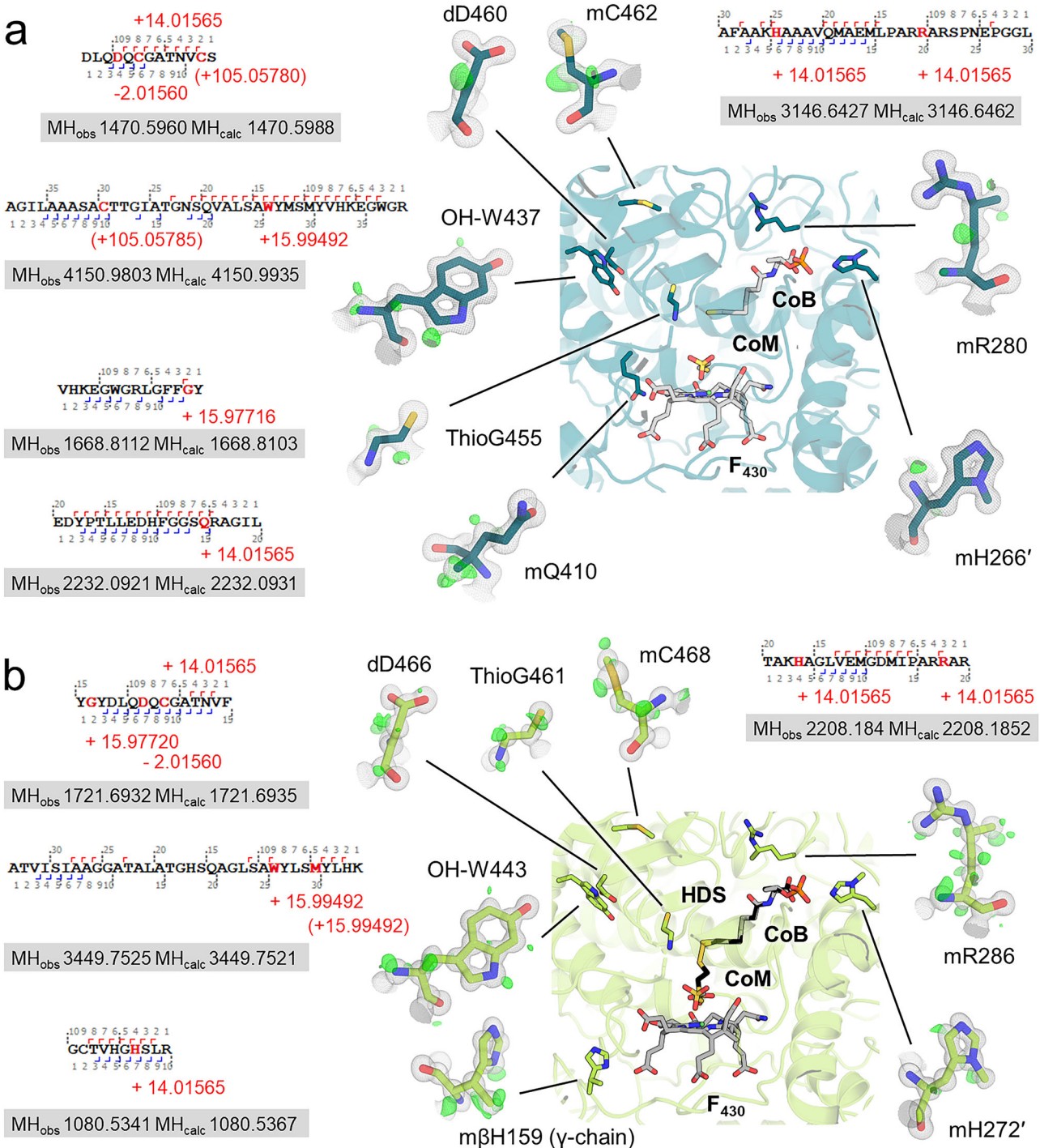

**Fig. 4 | ANME-2 MCR post-translational modifications.** PTMs in the active site of **a** ANME-2c and **b** ANME-2d$^V$. The main chain is shown as a cartoon, with ligands and PTMs shown as sticks with colours according to atoms (red for O, blue for N, orange for P, light green for Ni, and yellow for S). Close-ups of PTMs flank the central picture, highlighting electron densities. The $2F_o-F_c$ map (grey mesh) is contoured at 2σ, and the $F_o-F_c$ map (green mesh) is contoured at 3σ. The latter highlights the experimental hydrogen positions. The 3(S)-methylhistidine is positioned on the γ-chain. PTMs are labelled with the modification-one-letter amino acid code-residue number. m stands for methyl-, d for didehydro-, OH for hydroxy-, mβ for beta-methyl. A representative peptide sequence with fragment ions (hooks) is shown for each PTM. Mass shifts are highlighted in red and shown in brackets if artefactual. The observed (MH$_{obs}$) and calculated (MH$_{calc}$) monoisotopic mass are shown in a box below. Chymotrypsin was used to detect the 6-hydroxytryptophan, and all others with trypsin.

the electron density map, all PTMs could also be confirmed via LC-MS analysis (Fig. 4 and Supplementary Figs. S16–18).

## Discussion

ANME have a significant impact on the global carbon cycle due to their important role as a biological methane filter. The efficiency of their catabolism could be harnessed in microbial-driven technology, as illustrated by electrode-attached ANME-2d producing an electrical current from methane oxidation[14] or as a methane sink in wastewater[39]. Therefore, efforts are carried out by the scientific community to decipher their metabolisms. Here, by directly exploiting microbial consortia, we isolated and structurally characterised the key catabolic

enzymes from ANME representatives, expanding the available pool of methanotrophic MCR structural models from one to four. Due to their excellent quality, the MCR crystals from ANME-2d provide subatomic resolution, detailing the active site, cofactor, and coenzymes to unprecedented levels. Further structural studies using these models could be conducted to assign protonation states through neutron diffraction or time-resolved crystallography, facilitated by the rapid micro-crystal formation and the high protein yields. This presents opportunities for investigating the mechanism of methane activation.

Compared to active MCR retrieved from hydrogenotrophic methanogens, it is currently not feasible to obtain reactivated ANME enzymes with an in vivo approach. Still, our structural insights showed that methane-oxidising MCRs do not contain gas tunnels that connect the solvent with the buried active site, in contrast to the ethane-activating enzyme. Our structural analysis suggests that methane first enters the catalytic site before the heterodisulfide closes the catalytic chamber[19]. Such a mechanism is likely to be shared among all ANME MCRs.

Structural differences observed between MCRs from ANME-2 and ANME-1 are more likely rooted in their distinct evolutionary histories than in adaptations to specific ecological niches. This is supported by our finding that MCRs from both freshwater and marine ANME-2 archaea are highly conserved despite originating from different environments. According to proposed evolutionary scenarios, methanotrophy in ANME-1 likely evolved from alkanotrophic ancestors (most likely Alkanophaga, however, with an MCR that would originate from *Methanofastidiosa*), while in ANME-2, it emerged from methanogenic lineages (*Methanosarcinales*)[40]. These differences suggest that the MCR architectures in ANME-1 and ANME-2 arose through convergent evolution, rather than from a shared methanotrophic ancestor. Beyond differences in cofactors (e.g. methylthio-$F_{430}$ versus classic $F_{430}$) and structural particularities (e.g. presence or absence of a cysteine-rich patch), the repertoire of PTMs is largely divergent between clades (Fig. 4 and Supplementary Fig. S14). However, three structural positions appear consistently important for methanotrophy: a conserved $N^1$-methylhistidine, a thioglycine, and a 5(*S*)-methylarginine in ANME-2/methanogens compensated by S-oxymethionine/7-hydroxytryptophan in ANME-1.

With seven PTMs, ANME-2 MCRs gather the largest repertoire of modifications reported among methane-metabolising archaea. Notably, the structurally closest related MCR from *M. shengliensis* (also belonging to the same order[41]) harbours the most reduced gallery with three PTMs[35]. A rational explanation for the great number of PTMs in ANME would be the kinetic limitation of methane activation compared to the methane release reaction. The methane oxidation in MCR has an app$K_m$ value of ≈10 mM (measured with the MCR isoform I of *M. marburgensis*). This strongly limits the catalytic rates of methane activation by ANME at in situ conditions, where methane concentrations are often below 1 mM. Recent studies suggest that PTMs stabilise the active site and ligands[25,26], so it might be beneficial for a microbe to invest in equipping its core catabolic machinery to enhance, even slightly, its kinetic properties (i.e. by further decreasing the app$K_m$). Further studies of other ANME MCRs will be necessary to observe a correlation between the increased amounts of PTMs and methanotrophy. Similarly, future studies could investigate the prevalence of the 3(*S*)-methylhistidine and whether it exists in other *Methanosarcinales*. To our knowledge, this is the first report of a 3(*S*)-methylhistidine in a natural protein and the first PTM identified on the γ-chain of any characterised MCR. While methylation occurs on a residue coordinating the $F_{430}$ cofactor, it appears to have no immediate effect on the surrounding active site, raising questions about its role, which is in line with the so far mostly unresolved functions of the majority of PTMs observed in MCRs[19,28]. The location of the 3(*S*)-methylhistidine interacting directly with $F_{430}$ suggests a compensation for the lack of the methylglutamine in ANME-2d$^O$ and ANME-2d$^V$ placed in a close

proximity (Supplementary Fig. S15). Because all crystallographic structures are mere snapshots of a single state, we cannot exclude a role of the 3(*S*)-methylhistidine and methylglutamine during catalysis, enzyme assembly, or $F_{430}$ reactivation. Located on a loop at the surface of the γ-subunit, the 3(*S*)-methylhistidine could be installed before the oligomeric assembly by a dedicated methyltransferase. The only known installation of a methyl group on a Cβ in MCR is from the '*Ca*. E. thermophilum' homologue in which an isoleucine from the alpha subunit is modified to generate a hydrophobic gas channel. Since isoleucine and histidine have drastic chemical differences, the enzyme responsible for the modification should not be the same. Due to the low nucleophilicity of the β-carbon, the reaction should occur similarly to the installation of a methylarginine (installed by the Mmp10 methyltransferase[26]) or methylglutamine[27] rather than the $N^1$-methyl-histidine, in which an S-adenosylmethionine-dependent methyl-transferase would substitute a hydrogen of the imidazole ring nitrogen with a methyl group.

By revealing the unexpected features of methanotrophic MCRs through accurate models, our work emphasises the need to understand how native systems operate. Highlighting which type of cofactor should be inserted or which PTMs would be appropriate to install would guide the design of efficient heterologous methane capture systems for mitigating greenhouse gas emissions and moving towards a zero-carbon emission society.

## Methods

### Biomass acquisition and microbial enrichments

**ANME-2c cultivation.** Sediment was sampled using a multicorer at Amon mud volcano, Nile deep-sea fan, Mediterranean Sea at 032° 21.70 N, 031° 23.40 E at a water depth of 1120 m and 14 °C during NAUTINIL cruise in 2003 with the research vessel RV L'Atalante. Onboard samples were transferred into gas-tight bottles. In the home laboratory, sediment slurries were set up with a synthetic marine sulfate reducer medium at pH 7.4[42] and methane headspace at 20 °C. Incubated slurries showed methane-dependent sulfate reduction within a few days. To remove sediment debris, cultures were continuously incubated with methane as the sole electron donor and sulfate as the electron acceptor. The activity-based doubling time for both cultures, monitored by sulfide measurements, was about 4 to 6 months. Consecutive dilution and further incubation resulted in debris-free cultures of detached flocks essentially composed of AOM-performing methane-oxidising archaea and sulfate-reducing bacteria consortia. For more details on the cultivation methods, see Laso-Pérez et al.[43].

**ANME-2d$^O$ bioreactor cultivation of Ooijpolder sample.** Granular biomass was obtained from a highly enriched bioreactor culture[33]. The culture was maintained in an anaerobic 15 L sequencing fed-batch reactor (30 °C, pH 7.3 ± 0.1 controlled with KHCO$_3$, stirred at 200 rpm, with a working volume between 8 and 11 L). The reactor was continuously sparged with 15 mL min$^{-1}$ CH$_4$/CO$_2$ (95:5) and fed with medium (flow rate 2–2.5 L day$^{-1}$), as described in Kurth et al.[32] with an adaptation of the KH$_2$PO$_4$ concentration to 0.05 g L$^{-1}$ and sodium nitrate (15–25 mM) as electron acceptor. Once a day, the biomass was settled for 5 min, after which the excess liquid in the reactor was pumped off to a volume of 8 L.

**ANME-2d$^V$ bioreactor cultivation of Vercelli rice fields sample.** Granular biomass was obtained from a freshwater bioreactor enrichment culture first described by Vaksmaa et al.[17]. The culture was maintained in an anaerobic 15 L sequencing-fed batch reactor (room temperature, pH 7.3 ± 0.1 controlled with KHCO$_3$, stirred at 200 rpm) with a working volume between 8 and 11 L. The reactor was continuously sparged with 10 mL min$^{-1}$ CH$_4$/CO$_2$ (95:5) and fed with medium (flow rate 1.5–2 L day$^{-1}$) containing sodium nitrate (5–8 mM).

Once a day, the biomass was settled for 5 min, after which the excess liquid in the reactor was pumped off to a volume of 8 L.

The medium contained 0.07 g L$^{-1}$ MgSO$_4$·7H$_2$O, 0.1 g L$^{-1}$ CaCl$_2$·2H$_2$O, 0.05 g L$^{-1}$ K$_2$HPO$_4$, 0.5 mL L$^{-1}$ of trace element solution, 0.3 mL iron solution (54 g L$^{-1}$ FeCl$_3$·6H$_2$O and 114.6 g L$^{-1}$ nitrilotriacetic acid), and 0.1 mL L$^{-1}$ of Wolin's vitamin solution (DSMZ 141). The trace element solution contained per liter: 0.12 g CeCl$_3$·7H$_2$O, 0.6 g CoCl$_2$·6H$_2$O, 4.0 g CuSO$_4$, 0.07 g H$_3$BO$_3$, 1.0 g MnCl$_2$·4H$_2$O, 0.35 g Na$_3$MoO$_4$·2H$_2$O, 0.1 g Na$_2$WO$_4$·2H$_2$O, 0.95 g NiCl$_2$·6H$_2$O, 0.14 g SeO$_2$, 1.44 g ZnSO$_4$·7H$_2$O.

### Metagenome sequencing

**ANME-2c.** For DNA extraction, 20 ml of the Amon mud volcano culture was sampled and pelleted by centrifugation. From the pellet, microbial DNA was extracted using the DNeasy PowerSoil Kit (Qiagen, Venlo, Netherlands) following the supplied protocol. Library preparation and Illumina sequencing (2 × 150 bp reads, 150 million raw reads) were performed at LGC Genomics (Berlin). The metagenomic reads were assembled using SPAdes (v. 4.0.0) with default parameters[44] and automatically binned with the software SemiBin (v. 2.1.0) on the de novo assembly using the *single_easy_bin* pipeline[45]. Three bins affiliated with ANME-2 were obtained according to GTDB (v. 2.1.1) classification[46]. Annotation was carried out with Prokka (v. 1.14.6)[47], followed by extraction of the corresponding MCR proteins.

**ANME-2d.** Genomes were retrieved from GTDB-Tk (v. 2.4.0) for ANME-2d$^O$, '*Ca.* Methanoperedens sp.' BLZ2 under accession number GCA_002487355.1 and ANME-2d$^V$, '*Ca.* Methanoperedens' Vercelli Strain 1 under accession number GCA_905339155.1. MCR sequences from '*Ca.* Methanoperedens sp.' BLZ2 correspond to NCBI entries WP_097300250.1, WP_097300253.1, and WP_097300251.1 for the α, β, and γ subunits, respectively. MCR sequences from '*Ca.* Methanoperedens' Vercelli strain 1 correspond to NCBI entries MCX9030096.1, MCX9030093.1, and MCX9030095.1, for the α, β, and γ subunits, respectively.

DNA extraction and sequencing of ANME-2d$^O$ and ANME-2d$^V$ were performed as described previously[48]. Microbial taxonomic profiles of the different ANME enrichments were determined using SingleM[49] against the GTDB-Tk database (v. 2.4.0). Read quality was assessed using FASTQC (v. 0.11.9). The ANME-2c sample consisted of 83 million reads of 150 bp, which resulted in a 2367× profile coverage, while the ANME-2d$^O$ and ANME-2d$^V$ samples consisted of 6.6 and 5.5 million reads of 300 bp, respectively, resulting in 572× and 574× profile coverage.

### Sample treatment and lysis

**ANME-2c.** Two anaerobic 150 mL AOM enrichment cultures were used as starting material. Particles were settled overnight at RT, followed by the removal of ~130 mL of clear supernatant. H$_2$S in the culture headspace was replaced with N$_2$ gas before transferring the culture to an anaerobic chamber. Cell lysis and preparation of extracts were performed in a N$_2$/CO$_2$ atmosphere (90:10%) at 31 °C. Both samples were pooled and centrifuged at 8000 × $g$ for 5 min. The supernatant was carefully removed, resulting in a pellet of ~3.21 g (wet weight). The aggregates were resuspended in 30 mL IEC-A buffer (50 mM Tricine/NaOH pH 8 and 2 mM dithiothreitol (DTT)) and lysed via two rounds of sonication interceded with a French Press treatment at 1000 PSI (6.895 MPa). The French press cell was flushed with N$_2$ gas before use to ensure anoxic conditions. Unfortunately, 7.5 mL of the lysed sample was lost in the airlock due to an accidental break of the serum flask. The leftover total extract was then diluted to 60 mL with IEC-A, followed by centrifugation at 45,000 × $g$ for 30 min at 18 °C. A volume of 100 μL was withdrawn at each step for SDS-PAGE analysis (Supplementary Fig. S2).

**ANME-2d$^O$.** Frozen biomass containing '*Ca.* Methanoperedens sp.' BLZ2 (17.5 g of cells) was transferred to an anaerobic chamber containing an N$_2$/CO$_2$ atmosphere (90:10%) and resuspended in 90 mL buffer IEC-A. Cells were first lysed via two rounds of sonication (SONOPULSE Brandelin, 75% power, 30 s, 1st round: 4 cycles, 2nd round: 3 cycles) interceded with a French Press treatment at 1000 PSI (6.895 MPa). The French press cell was previously flushed with N$_2$ to ensure anoxic conditions. After the second round of sonication, the sample was centrifuged at 45,000 × $g$ for 30 min at 18 °C. The supernatant was collected and subjected to ultracentrifugation at 100,000 × $g$ for 1.5 h at 4 °C (rotor used: 70Ti, Beckman Coulter). A volume of 100 μL of the total cell extract and soluble fraction was withdrawn for the Bradford assay and SDS-PAGE.

**ANME-2d$^V$.** The frozen biomass (12.47 g) was resuspended in 60 mL buffer IEC-A. Cells were processed the same way as ANME-2d$^O$ except for the following variation. The supernatant from ultracentrifugation was sonicated for an additional four cycles, and 40 mL of filtered supernatant was obtained before injection onto the HiTrap Q HP column.

### MCR purification

**ANME-2c.** Protein purification was performed under anoxic conditions in an anaerobic Coy tent with an N$_2$/H$_2$ atmosphere (97:3%) at 20 °C under yellow light. Before every injection onto a chromatography column, the sample was filtered using a 0.2 μm nitrocellulose filter (Sartorius, Germany). Proteins were tracked via multi-wavelength absorbance monitoring (λ 280, 415, and 550 nm) in combination with SDS-PAGE. The filtered soluble extract (60 mL; 20.7 mg protein) was loaded onto a 1 mL HiTrap Q-Sepharose HP column (Cytiva, Munich, Germany) equilibrated with IEC-A. The proteins were eluted with a two-step protocol, reaching 0.5 M and then 1 M at a flow rate of 0.7 mL min$^{-1}$. MCR was eluted at 0.5 M NaCl, and the pool (-13 mL) was diluted ~1:4 (protein sample: buffer) with IEC-A buffer. The filtered sample was loaded on a 0.982 mL Mono Q 5/50 column (Cytiva, Munich, Germany). The elution was performed at a gradient of 100 to 700 mM NaCl at 0.5 mL min$^{-1}$. The pooled MCR fractions were diluted ~1:4 (protein sample: buffer) with HIC-B buffer (25 mM Tricine/NaOH pH 7.6, 2 M (NH$_4$)$_2$SO$_4$, and 2 mM DTT). The pool was filtered and passed on a 1.622 mL Source15φ column (Cytiva, Munich, Germany). Proteins were eluted by applying a gradient of 1.5 to 0 M (NH$_4$)$_2$SO$_4$ at 0.6 mL min$^{-1}$. The final MCR pool was washed by dilution (1:1000) with storage buffer (25 mM Tris/HCl pH 7.6, 10% v/v glycerol, and 2 mM DTT) using a 30 kDa cut-off concentrator to remove salts. Finally, the sample was concentrated to 200 μL. The protein concentration estimated via the Bradford method was 24.6 mg mL$^{-1}$.

**ANME-2d$^O$.** Purification was performed under the previously described conditions, and fractions of interest were followed at λ 280 nm. The soluble extract (90 mL at 7.82 mg mL$^{-1}$) could not be filtered due to the large particulate size and was directly injected into the 10 mL HiTrap Q-Sepharose HP column pre-equilibrated with IEC-A buffer. After washing the column, the elution was performed with a gradient of 0 to 650 mM NaCl for 10 column volumes (CV) at a 2 mL min$^{-1}$ flow rate. The pool of interest was diluted 1:1 with HIC-B buffer, filtered, and loaded on the 1.622 mL Source 15φ column. A gradient from 80% to 0% (NH$_4$)$_2$SO$_4$ for 10 CV at a 1.5 mL min$^{-1}$ flow rate was applied. The final MCR fractions were pooled and concentrated on a Vivaspin centrifugation concentrator with a cut-off of 30 kDa. The buffer was exchanged by ultrafiltration for the storage buffer (25 mM Tris/HCl pH 8.0, 10% v/v glycerol, 2 mM DTT, and 100 mM NaCl) containing NaCl to prevent the spontaneous formation of microcrystals. Around 2 mL of purified MCR was recovered at 57 mg mL$^{-1}$, aliquoted into 50 μL, and stored at 4 °C and −80 °C.

**ANME-2dᵛ.** Purification was performed under the previously described conditions, and fractions of interest were followed at λ 280 nm. The filtered soluble extract (40 mL) was injected into the 10 mL HiTrap Q HP column pre-equilibrated with IEC-A buffer. After washing the column, the elution was performed with a gradient of 0 to 650 mM NaCl for 16 CV at a 2 mL min⁻¹ flow rate. The pool of interest was diluted 1:2 (protein sample:buffer) with HIC-B buffer, filtered, and loaded on a 5 ml HP Phenyl Sepharose column. A gradient from 1.3 to 0 M of $(NH_4)_2SO_4$ for 12 CV at a 1.0 mL min⁻¹ flow rate was applied. The final MCR fractions were pooled and concentrated using a Vivaspin centrifugation concentrator with a 30 kDa cut-off. Ultrafiltration exchanged the buffer for storage buffer. Around 0.35 mL of purified MCR was recovered at 40 mg mL⁻¹ and stored at 4 °C and −80 °C.

## UV−visible spectrophotometry

UV−visible absorption spectra were recorded using a Cary 60 spectrophotometer (Agilent Technologies, Waldbronn, Germany) on a TrayCell with a 1 mm path length.

## Phylogeny analyses

The three MCR subunits from each MAG were concatenated and used to calculate a phylogenetic tree, including the sequences of the studied MCRs from ANME-2c and ANME-2d of this manuscript and the sequences of structural homologues. The concatenated MCR sequences were aligned using MUSCLE (v. 3.8.1551)[50]. The aligned concatenated sequences were used to calculate a phylogenetic tree with RAxML (v. 8.2.12)[51] using a partition file to calculate differential models for each gene and the following parameters '-m PROTGAMMAAUTO -f a -N autoMRE -k'.

## Crystallisation

**ANME-2c.** The fresh preparation of MCR was immediately crystallised. Crystals were obtained using the sitting drop method on 96-Well MRC 2-Drop Crystallisation Plates in polystyrene (SWISSCI, Switzerland), containing 90 μL of crystallisation solution in the reservoir. 0.5 μL protein sample at a concentration of 24.6 mg mL⁻¹ was mixed with 0.5 μL of crystallisation solution. The crystallisation screening was performed by an OryxNano (Douglas Instrument, UK) at 18 °C in a tent filled with 100% $N_2$. Then, plates were transferred to a Coy tent ($N_2$/$H_2$ atmosphere 97:3%) at 20 °C. Crystals were formed over the following days. The most successful conditions were: (A) 30% v/v 2-Methyl-2,4-pentanediol, 100 mM Tris pH 8.5, 500 mM NaCl and 8% w/v Polyethylene glycol 8000, and (B) 30% v/v Polyethylene glycol 400, 100 mM HEPES pH 7.5, 200 mM $MgCl_2$.

The conditions were replicated on a Junior Clover plate (Jena Bioscience, Germany) under oxic conditions with 90 μL of the crystallisation solution in the reservoir. Crystals with yellow square or rectangular morphology appeared over the next two weeks. The final dataset of ANME-2c MCR was obtained from crystals formed from 2 μL of protein and 2 μL of condition A.

**ANME-2dᵒ.** Following an initial screening on a 96-well MRC 2-Drop Crystallisation Plate done aerobically at 18 °C, MCR crystals were reproduced using the sitting drop method on a Junior Clover plate. The best crystals were obtained with a reservoir filled with 100 μL of the following crystallisation solution: 20% w/v polyethylene glycol 3350, 50 mM Tris pH 8.0 and 200 mM potassium nitrate and by spotting 5 μL protein at a concentration of 2.19 mg mL⁻¹ with 2 μL crystallisation solution. Thick yellow brick-shaped crystals appeared within 24−48 h.

**ANME-2dᵛ.** Following an initial screening on a 96-well MRC 2-Drop Crystallisation Plate done aerobically at 18 °C, MCR crystals were reproduced using the sitting drop method on the Junior Clover plate. Here, the reservoir was filled with 100 μl of 20% w/v Polyethylene glycol 3350 and 200 mM sodium thiocyanate pH 6.9, and the crystallisation

well contained 3 μL protein at a concentration of 40 mg mL⁻¹ with 2 μL mother liquor.

## Krypton labelling

ANME-2c MCR in solution was shipped to the High-Pressure Freezing Laboratory for Macromolecular Crystallography (HPMX)[52] of the European Synchrotron Radiation Facility (ESRF, Grenoble, France). Crystallisation was repeated using the crystallisation solution from condition B (see crystallisation of ANME-2c MCR) at ambient temperature in an aerobic environment on a Junior Clover plate (Jena Bioscience, Germany). Obtained crystals were used for krypton-labelling using the pressurisation method. Here, crystals were pressurised at 214 bars of krypton for 2 min prior to being flash frozen at cryogenic temperature, still under pressure. X-ray crystallographic data were collected on the beamline ID23-1 (ESRF) at 14.4 keV, above the krypton absorption edge.

## X-ray data collection, processing, modelling, and validation

MCR crystals were soaked in the crystallisation solution supplemented with 20% v/v ethylene glycol for ANME-2dᵒ and ANME-2dᵛ for 5−10 s before being transferred to liquid nitrogen.

X-ray diffraction for ANME-2c crystals was collected at SOLEIL (Source optimisée de lumière d'énergie intermédiaire du LURE) synchrotron, Saint-Aubin, France, at the beamline PROXIMA-1. Crystals for both ANME-2d were collected at the ESRF, Grenoble, France, on the beamline BM07-FIP2. Krypton-derivative ANME-2c crystals were collected at the ESRF on ID23-1 beamline.

The datasets were processed and scaled with *autoPROC* (v. 1.0.5, Global Phasing Limited, Cambridge, UK)[53], except for the krypton-derivative sample, which was integrated with XDS and scaled with *aimless* (v. 1.12.14) from the CCP4 package. The structures were solved by molecular replacement using *PHASER* from the *PHENIX* package (v. 1.19.2-4158)[54]. For molecular replacement, we used an AlphaFold 2[55] model based on the metagenome-derived amino acid sequence for MCR of ANME-2c. The ANME-2c MCR model was used as a template to solve krypton-derivative MCR ANME-2c and as-isolated ANME-2dᵒ MCR datasets. Finally, the ANME-2dᵒ MCR model was used as a template to solve ANME-2dᵛ MCR.

All models were then refined with *COOT* (v. 0.8.9.2)[56] and *PHENIX* refine (v. 1.19.2-4158). Because of the high resolution, all atoms were considered anisotropic (except for the krypton-derivative model), and hydrogens were added in riding mode. ANME-2dᵛ MCR was refined with a mix of ligands in the active site, since this resulted in the best fit to the electron density (Supplementary Fig. S9). All models have been deposited with hydrogen atoms modelled on the protein (e.g. not on ligands and solvents).

The models were validated by the MolProbity server (http://molprobity.biochem.duke.edu). Figures of MCR structures were generated with PyMOL (v. 2.2.0, Schrödinger, LLC). Sequence alignments were generated with MAFFT 7[57] and visualised with ESPript 3.0[58]. Internal tunnels analyses connecting the active site to the surface were performed by CAVER with default parameters and considering a radius of 0.9[37]. With these parameters, no tunnels could be detected except for the ECR control.

Examples of local electron density and omit map for ligands can be found in Supplementary Figs. S19−23.

## Mass spectrometry analysis

Purified MCRs in solution were used for liquid chromatography mass spectrometry (LC-MS) analysis to confirm the presence of the PTMs observed in the electron density.

The purified proteins were digested after carbamidomethylation with trypsin, chymotrypsin, or elastase at 30 °C in ammonium bicarbonate buffer. In order to obtain many different peptides, sodium lauryl sulfate, sodium deoxycholate, or guanidine hydrochloride were

added in various digestion approaches. The duration of incubation with protease was varied from 1.5 to 15 h. In addition, ANME-2c MCR was labelled via vinylpyridine or iodoacetamide. The peptides were then desalted using C18 SPE and measured using LC-MS carried out on an Exploris 480 instrument connected to an Ultimate 3000 RSLC nano and a nanospray flex ion source (all ThermoFisher Scientific, USA). The separating gradient of 60 min ran with solvent A, 0.15% formic acid in water, and solvent B, 0.15% formic acid in acetonitrile. The data were recorded in DDA mode and analysed using the Byonic or Proteome-discoverer software and the sequence data files (FASTA).

It should be noted that the hydroxytryptophan could not be found reliably with Byonic and often included artificial new adduct masses. Additional analysis with Proteomediscoverer (Sequest) has identified the modified residue reliably in both ANME-2d MCRs.

The described data can be found in the following link: https://zenodo.org/records/15784324.

### Reporting summary

Further information on research design is available in the Nature Portfolio Reporting Summary linked to this article.

## Data availability

The metagenomic sequences from the Amon mud volcano enrichment generated in this study were deposited in the NCBI BioProject database under accession code PRJNA1248948. The metagenomic sequences for both ANME-2d samples were deposited in the NCBI BioProject database under accession code PRJNA1289043. The structural data generated in this study (structure factors and models) have been deposited in the Protein Data Bank under the accession codes 9QR3 (as isolated MCR ANME-2c, https://doi.org/10.2210/pdb9QR3/pdb), 9QM5 (Krypton-pressurised MCR ANME-2c, https://doi.org/10.2210/pdb9QM5/pdb), 9QR1 (MCR ANME-2d$^O$, https://doi.org/10.2210/pdb9QR1/pdb), and 9QQT (MCR ANME-2d$^V$, https://doi.org/10.2210/pdb9QQT/pdb). The mass spectrometry data generated in this study have been deposited on the Zenodo server under accession record 15784324 (doi 10.5281/zenodo.15784324). The SDS-PAGE/hrCN-PAGE data generated in this study are provided in the Source Data file. The UV/Vis data generated in this study are provided in the Source Data file. Source data are provided with this paper.

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

## Acknowledgements

We are thankful to the Max Planck Institute for Marine Microbiology and the Max Planck Society for their continuous support. We thank Christina Probian and Ramona Appel of the Microbial Metabolism laboratory for the technical assistance. We acknowledge the French Biology/Health Panel Review Committee for the provision of synchrotron radiation beamtime at SOLEIL (Saint Aubin, France) on beamline PROXIMA-1 and at the ESRF (Grenoble, France) on beamline BM07-FIP2. BM07-FIP2 is supported by the French ANR PIA3 (France 2030) EquipEx+ project MAGNIFIX under grant agreement ANR-21-ESRE-0011. We acknowledge the ESRF for providing synchrotron radiation beamtime on beamline ID23-1 and access to the HPMX laboratory. T.W. was supported by a European Research Council consolidator grant [Grant number 101125699]. The initial crystallisation screening performed by an Oryx-Nano robot was financed by the Deutsche Forschungsgemeinschaft (DFG) priority programme 1927 Iron-Sulphur for Life [Grant number WA 4053/1-1]. C.U.W. thankfully acknowledges financial support from the Dutch Science Foundation through an NWO-VIDI Talent Grant [Grant number VI.Vidi.223.012] and the SIAM Gravitation Grant [Grant number 024.002.002]. R.L.-P. was funded by a Ramón y Cajal grant (RyC2021-031775-I) from the Spanish Ministerio de Ciencia e Innovación (MCIN/AEI/10.13039/501100011033) and the European Union («NextGenerationEU»/PRTR).

## Author contributions

Enrichment of ANME-2c was done by G.W., enrichments of ANME-2d were done by M.W. and C.U.W. Metagenomic data processing for ANME-2c and ANME-2d was done by R.L.-P. and M.W., respectively. Purification and crystallisation of ANME-2c MCR were performed by M.-C.M. and N.V.P. Purification and crystallisation of ANME-2d$^O$ MCR were done by P.M. and M.W. Purification and crystallisation of ANME-2d$^V$ were done by M.W. and T.W. Data collection was done by S.E. and T.W. Krypton pressurisation was done by S.E. and P.C. X-ray data processing, model refinement and analysis were done by M.-C.M. and model validation by T.W. LC-MS processing and analyses were performed by J.K. The

manuscript was written by M.-C.M. and T.W. Manuscript edition and correction were done by all authors.

## Funding

## Competing interests
The authors declare no competing interests.

## Additional information

[1]Max-Planck-Institute for Marine Microbiology, Bremen, Germany. [2]Department of Microbiology, Radboud Institute for Biological and Environmental Sciences (RIBES), Radboud University, Nijmegen, The Netherlands. [3]Biogeochemistry and Microbial Ecology Department, Museo Nacional de Ciencias Naturales (MNCN-CSIC), Madrid, Spain. [4]Univ. Grenoble Alpes, CEA, CNRS, Institut de Biologie Structurale, Grenoble, France. [5]European Synchrotron Radiation Facility, Grenoble, France. [6]Univ. Grenoble Alpes, CEA, CNRS, IRIG-LCBM UMR 5249, Grenoble, France. [7]Max Planck Institute for Terrestrial Microbiology, Marburg, Germany. [8]MARUM, Center for Marine Environmental Sciences, University of Bremen, Bremen, Germany. ✉e-mail: twagner@mpi-bremen.de

