## [Transparent Peer Review file · Nature Communications]

Atomic resolution structures of the methane-activating enzyme in anaerobic methanotrophy reveal extensive post-translational modifications

Corresponding Author: Dr Tristan Wagner

Version 0:

Reviewer comments:

Reviewer #1

(Remarks to the Author)

The manuscript from Muller et al. reports the in situ purification of methyl coenzyme M reductases (MCRs) from anaerobic methanotrophic archaea (ANME) of both freshwater and marine origins. The study describes the crystal structures of these MCR at atomic resolution revealing structural conservation with the methanogenic MCRs rather than the phylogenetically divergent MCR that has previously been reported. The high resolution of the structures allows for detailed characterization of the post-translational modifications that are found in each of the MCRs including a novel 3(S)-methylhistine modification in the ANME-2d MCR. Lastly, gas labelling experiments did not reveal an obvious channel for alkane diffusion such as ones observed in ethane-specific MCR suggesting that the ANME MCRs are mechanistically similar to the methanogenic MCRs.

The study is a technically tour de force and not without many shares of challenges. That said, it is somewhat disappointing that there aren't any obvious signature features that distinguishes the structure of ANME MCRs from those of the methanogenic MCRs. The report of the methylation of the histine residue coordinating the F430 cofactor is exciting but is not really elaborate in detail in the results or discussion section. The authors should consider expanding on this finding. Lastly, a minor point that the mass spectrometric analysis describing the identification of the new post-translational modification should be included as a figure in the main text.

In short, this is new and exciting work that is above the bar for Nature Comm. provided that the point above are addressed in sufficient detail.

Reviewer #2

(Remarks to the Author)

In the manuscript „Atomic resolution structures of the methane-activating enzyme in anaerobic methanotrophy reveal extensive post-translational modifications“ Müller et al report the structures of three methyl-coenzyme M reductases (MCR) from different anaerobic methanotrophic archaea (ANME) belonging to the ANME-2d and ANME-2c groups. The MCRs from ANME catalyze the first step in anaerobic oxidation of methane (AOM), which represents the reverse reaction of methane formation catalyzed by MCRs from methanogenic archaea. So far, several structures of methanogenic MCRs as well as one structure of an ANME MCR (ANME-1) were solved. One characteristic feature of these enzymes is the high number of unusual posttranslational amino acid modifications (PTM).

In this work, the native proteins were purified from enrichment cultures, crystallized and the structures solved at atomic resolution. Overall, the structures are highly similar to those of previously investigated MCRs from methanogenic archaea. This includes the presence of an unmodified coenzyme F430, in contrast to the thiomethylated F430 in the MCR from ANME-1. Interestingly, seven PTMs were identified, among them a novel C-methylated histidine residue. These results clearly show the importance of native protein purification to unravel novel intrinsic enzyme features.

The presented work is of very high quality with respect to the experimental design and results. It provides important new insights into the enzymology of ANME. The paper is very well written and highly recommended for publication after minor revision (see below).

There are only a few minor comments:

1. Figure 1: For completeness, it would be nice to also see a picture of the crystals of the ANME-2dV MCR.
2. Figure 2: a) The ligands in the overall structure in gray are hardly visible and could be omitted. b) N- and C-termini are both shown as spheres. It would be better to use spheres for one terminus and another symbol for the other terminus.
3. Figure 3: Although it is mentioned in the figure legend that only the interacting side chains are shown, it is a little bit confusing that certain side chains are completely omitted. For example, the alpha-MY346-347 in Figure 3b looks like a GY sequence. I would suggest to show all side chains of the shown residues and only omit non-interacting main chains.
4. Figure 4: In this figure, the three-letter code for amino acid residues is used, in Figure 3 it is the one-letter code. For the sake of consistency, please use either one or the other. (also in Supplemental Figure S13)
5. Supplemental Figure S2: Please show SDS-PAGE analysis and UV/Vis absorption spectra (close up and full spectrum) for all three purified MCRs.
6. Supplemental Figure S14: Please label PTMs with either one- or three-letter standard code (see also comment 4).

Reviewer #3

(Remarks to the Author)

Response to the reviewers

We thank the reviewers for their time, insightful comments, and suggestions. The point-by-point responses can be found below.

Reviewer #1 (Remarks to the Author)

The manuscript from Muller et al. reports the in situ purification of methyl coenzyme M reductases (MCRs) from anaerobic methanotrophic archaea (ANME) of both freshwater and marine origins. The study describes the crystal structures of these MCR at atomic resolution revealing structural conservation with the methanogenic MCRs rather than the phylogenetically divergent MCR that has previously been reported. The high resolution of the structures allows for detailed characterization of the post-translational modifications that are found in each of the MCRs including a novel 3(S)-methylhistine modification in the ANME-2d MCR. Lastly, gas labelling experiments did not reveal an obvious channel for alkane diffusion such as ones observed in ethane-specific MCR suggesting that the ANME MCRs are mechanistically similar to the methanogenic MCRs.

The study is a technically tour de force and not without many shares of challenges. That said, it is somewhat disappointing that there aren't any obvious signature features that distinguishes the structure of ANME MCRs from those of the methanogenic MCRs. The report of the methylation of the histine residue coordinating the F₄₃₀ cofactor is exciting but is not really elaborate in detail in the results or discussion section. The authors should consider expanding on this finding. Lastly, a minor point that the mass spectrometric analysis describing the identification of the new post-translational modification should be included as a figure in the main text.

In short, this is new and exciting work that is above the bar for Nature Comm. provided that the point above are addressed in sufficient detail.

We are delighted that the referee appreciated our work and pointed out the technical tour de force.

As recommended by the referee, we have now supplied additional information in the results and discussion regarding the 3(S)-methylhistidine. We also added a new panel in Supplementary Fig. S15 to precisely overlay the F₄₃₀ and compare the subtle reposition of the coordinated propionate group.

Results, lines 235-241: "Upon careful analysis of F₄₃₀ superposition based on atomic resolution structures (Supplementary Fig. S15), we noticed that the methylhistidine slightly repositions the close-by F₄₃₀ propionate group in front of an axial histidine, optimising the contact without modifying the salt bridge distance (2.9 Å). It must be noted that the modified histidine 159 is located on a loop that would be exposed to the solvent on the γ -subunit alone. Therefore, while histidine 159 would be hindered in the MCR

complex, it would be accessible to a dedicated methyl-transferase for the PTM installation when the γ -subunit is nascent from the ribosome.”

Discussion, lines 298-308. “The location of the 3(S)-methylhistidine interacting directly with F₄₃₀ suggests a compensation for the lack of the methylglutamine in ANME2d^O and ANME2^V placed in a close proximity (Supplementary Fig. S15). Because all crystallographic structures are mere snapshots of a single state, we cannot exclude a role of the 3(S)-methylhistidine and methylglutamine during catalysis, enzyme assembly, or F₄₃₀ reactivation. Located on a loop at the surface of the γ -subunit, the 3(S)-methylhistidine could be installed before the oligomeric assembly by a dedicated methyltransferase. The only known installation of a methyl group on a C β in MCR is from the *Ca. E. thermophilum* homologue in which an isoleucine from the alpha subunit is modified to generate a hydrophobic gas channel. Since isoleucine and histidine have drastic chemical differences, the enzyme responsible for the modification should not be the same.”

Mass spectrometry analysis results have been integrated into the main Fig. 4 and Supplementary Figure S13 as advised. We retained the close-up and peptide coverage in Supplementary Figs. S16-18 to provide as much information as possible.

Reviewer #2 (Remarks to the Author)

In the manuscript „Atomic resolution structures of the methane-activating enzyme in anaerobic methanotrophy reveal extensive post-translational modifications“ Müller et al report the structures of three methyl-coenzyme M reductases (MCR) from different anaerobic methanotrophic archaea (ANME) belonging to the ANME-2d and ANME-2c groups. The MCRs from ANME catalyze the first step in anaerobic oxidation of methane (AOM), which represents the reverse reaction of methane formation catalyzed by MCRs from methanogenic archaea. So far, several structures of methanogenic MCRs as well as one structure of an ANME MCR (ANME-1) were solved. One characteristic feature of these enzymes is the high number of unusual posttranslational amino acid modifications (PTM).

In this work, the native proteins were purified from enrichment cultures, crystallized and the structures solved at atomic resolution. Overall, the structures are highly similar to those of previously investigated MCRs from methanogenic archaea. This includes the presence of an unmodified coenzyme F₄₃₀, in contrast to the thiomethylated F₄₃₀ in the MCR from ANME-1. Interestingly, seven PTMs were identified, among them a novel C-methylated histidine residue. These results clearly show the importance of native protein purification to unravel novel intrinsic enzyme features. The presented work is of very high quality with respect to the experimental design and

results. It provides important new insights into the enzymology of ANME. The paper is very well written and highly recommended for publication after minor revision (see below).

We would like to thank the referee for appreciating the importance and quality of our work.

There are only a few minor comments:

1. Figure 1: For completeness, it would be nice to also see a picture of the crystals of the ANME-2dV MCR.

An image of crystals of ANME-2dV was added to Figure 1.

2. Figure 2: a) The ligands in the overall structure in gray are hardly visible and could be omitted. b) N- and C-termini are both shown as spheres. It would be better to use spheres for one terminus and another symbol for the other terminus.

The ligands have been removed as advised in Figure 2.

For the N- and C-termini of Figure 2, the revised version has added labels to each close-up panel to avoid any ambiguity.

3. Figure 3: Although it is mentioned in the figure legend that only the interacting side chains are shown, it is a little bit confusing that certain side chains are completely omitted. For example, the alpha-MY346-347 in Figure 3b looks like a GY sequence. I would suggest to show all side chains of the shown residues and only omit non-interacting main chains.

Figure 3, together with Supplementary Figures S10 and S13, now includes all side chains in the revised version.

4. Figure 4: In this figure, the three-letter code for amino acid residues is used, in Figure 3 it is the one-letter code. For the sake of consistency, please use either one or the other. (also in Supplemental Figure S13)

All figures from the manuscript main and supplementary now contains the one letter code.

5. Supplemental Figure S2: Please show SDS-PAGE analysis and UV/Vis absorption spectra (close up and full spectrum) for all three purified MCRs.

Supplementary Figure S2 was expanded to include the SDS-PAGE purification profiles of ANME-2d^O and ANME-2d^V. Additionally, UV/Vis spectra of ANME-2d^O and ANME-2d^V were added and compiled in additional panels. The spectrum of *M. shengliensis* was removed due to the lack of replicates.

6. Supplemental Figure S14: Please label PTMs with either one- or three-letter standard code (see also comment 4).

Figure S14

Figure S14 has been revised as suggested.

Reviewer #3 (Remarks to the Author)

We are delighted that an Early Career Researcher had the opportunity of reviewing our manuscript.